# Customizable Collagen Vitrigel Membranes and Preliminary Results in Corneal Engineering

**DOI:** 10.3390/polym14173556

**Published:** 2022-08-29

**Authors:** María Dolores Montalvo-Parra, Wendy Ortega-Lara, Denise Loya-García, Andrés Bustamante-Arias, Guillermo-Isaac Guerrero-Ramírez, Cesar E. Calzada-Rodríguez, Guiomar Farid Torres-Guerrero, Betsabé Hernández-Sedas, Italia Tatnaí Cárdenas-Rodríguez, Sergio E. Guevara-Quintanilla, Marcelo Salán-Gomez, Miguel Ángel Hernández-Delgado, Salvador Garza-González, Mayra G. Gamboa-Quintanilla, Luis Guillermo Villagómez-Valdez, Judith Zavala, Jorge E. Valdez-García

**Affiliations:** 1Tecnologico de Monterrey, Escuela de Ingenieria, 2501 Garza Sada Ave., Colonia Tecnologico. C.P., 64849 Monterrey, NL, Mexico; 2Tecnologico de Monterrey, Escuela de Medicina, 3000 Morones Prieto Ave., Colonia Los Doctores. C.P., 64710 Monterrey, NL, Mexico

**Keywords:** scaffold, collagen vitrigel, tissue engineering, cornea, corneal endothelium

## Abstract

Corneal opacities are a leading cause of visual impairment that affect 4.2 million people annually. The current treatment is corneal transplantation, which is limited by tissue donor shortages. Corneal engineering aims to develop membranes that function as scaffolds in corneal cell transplantation. Here, we describe a method for producing transplantable corneal constructs based on a collagen vitrigel (CVM) membrane and corneal endothelial cells (CECs). The CVMs were produced using increasing volumes of collagen type I: 1X (2.8 μL/mm^2^), 2X, and 3X. The vitrification process was performed at 40% relative humidity (RH) and 40 °C using a matryoshka-like system consisting of a shaking-oven harboring a desiccator with a saturated K_2_CO_3_ solution. The CVMs were characterized via SEM microscopy, cell adherence, FTIR, and manipulation in an ex vivo model. A pilot transplantation of the CECs/CVM construct in rabbits was also carried out. The thickness of the CVMs was 3.65–7.2 µm. The transparency was superior to a human cornea (92.6% = 1X; 94% = 2X; 89.21% = 3X). SEM microscopy showed a homogenous surface and laminar organization. The cell concentration seeded over the CVM increased threefold with no significant difference between 1X, 2X, and 3X (*p* = 0.323). The 2X-CVM was suitable for surgical manipulation in the ex vivo model. Constructs using the CECs/2X-CVM promoted corneal transparency restoration.

## 1. Introduction

Tailoring biomaterials for corneal reconstruction via tissue engineering represents an important breakthrough that may alleviate the problem of tissue scarcity. The shortage of corneal graft tissue limits access to a transplant in ~53% of the world’s population. Although the cornea is the most frequently transplanted organ worldwide, there is currently only 1 cornea available for every 70 needed [1,2]. Given this situation, alternative solutions are needed to produce tissues for transplantation.

Corneal blindness is one of the four main causes of blindness around the world [3]. It can be caused by diseases, chemical burns, and trauma that affect different corneal layers [4]. The corneal endothelium is the innermost layer of the cornea. It regulates optimal corneal hydration to maintain the clarity needed for proper vision. This tissue does not regenerate and when the cell density decreases to a critical concentration, it produces an irreversible opacity that can only be treated with a corneal transplant [5]. Efforts in cell therapy have demonstrated the ability of corneal endothelial cells (CECs) to proliferate in vitro under the stimulus of supplemented culture media and retain their characteristic polygonal morphology and functional molecular markers, such as tight junctions (ZO-1) and Na-K/ATPase pumps [6,7,8]. However, given the monolayer nature of the corneal endothelium, CECs injected intracamerally do not always attach effectively to the corneal tissue [9].

Collagen membranes have the potential to serve as scaffolds for CECs in the production of bioengineered corneal tissue, given that type 1 collagen is an abundant protein in the corneal stroma and that it is produced by CECs [10,11,12,13]. These membranes are produced via a wide range of methodologies that vary in three main respects: (1) methodology, including desiccation, freeze drying, electrospinning, and 3D printing; (2) conditions, such as the collagen source and concentration, polymerization pH, and temperature; and (3) characterization, including microstructure, transmittance, FTIR, X-ray diffraction, cell adhesion, water uptake, biodegradability, and in vivo biocompatibility [14,15,16]. This leads to marked variations in parameters that limit the reproducibility of the methodologies and their application in corneal endothelium engineering [17].

Overall, the main challenges in producing collagen membranes for corneal endothelium engineering are the mechanical properties and biocompatibility. Enhancing the cross-linking of collagen fibers in the membranes is a strategy for the improvement of their mechanical properties while maintaining their low immunogenicity. Ultraviolet (UV) light treatment and desiccation in specific conditions are used for this purpose, producing vitrified membranes. Given that these strategies can produce transparent membranes, they have great potential for the restoration of cornea [18,19,20]. Currently, the protocols for producing collagen–vitrigel membranes (CVM) vary mainly in terms of temperature, time of desiccation, and relative humidity. This modifies the desired characteristics such as thickness, fiber diameter, density, and organization, which affect their application for clinical purposes.

The thickness of membranes is a parameter that is of special interest because it provides two main characteristics that determine their usefulness: surgical manipulation and cell adherence. A thicker collagen membrane is easier to surgically manipulate; however, this increases the distribution of charges, affecting the cell and tissue adherence [21]. A thinner membrane promotes cell and tissue adhesion but can be difficult to manipulate during surgery. For this reason, some collagen membranes reported for corneal engineering are more suitable for the epithelium or stromal layer, in which higher thickness provides the structure and barrier functions that cover different needs. 

In this research, we provide a simple and reproducible method to produce CVMs that can be tailored to assemble membranes with different characteristics with potential use in the engineering of different layers of the cornea. We aim to close the gap between the reported methods, leading to the development of standardized protocols that yield reproducible and quantitative characterization, which in turn facilitate comparative research and the development of prototypes for clinical applications [14,15,16,17,18,19,20,22,23,24,25,26,27,28,29,30,31].

## 2. Methods

### 2.1. Matryoshka System Assembly for the Production of Collagen Membranes 

For the manufacturing of CVMs, a matryoshka system was assembled (Figure 1).

It consisted of an incubator with a steady temperature (T) bearing a desiccator with a saturated K_2_CO_3_ solution to control the RH. For this purpose, 50 mL of K_2_CO_3_ solution were prepared using 1.15 g of K_2_CO_3_ (Sigma-Aldrich, P5833, St. Louis, MO, USA) per milliliter of bi distilled water. The salt solution was placed in a 200 mL beaker and fixed onto a 330 × 246 × 262 mm desiccator with a clear plastic dome (Thermo Scientific Nalgene, 5310-0250, Cleveland, OH, USA). A closed system was created by placing the sealed desiccator inside a shaking-plate incubator (I5211DS; Labnet International, Edison, NJ, USA). The T and shaking speed of the incubator were set at 40 °C and 30 rpm, respectively. The shaker function was turned on 48 h after the gel was placed inside the matryoshka system to avoid gel tilting. The T and RH were measured with a Monitoring Traceable Hygrometer (4040CC; Traceable^®®^ Products, Webster, TX, USA) placed inside the desiccator. The T and RH were registered daily until a temperature of 40 °C and RH of 40% were recorded consistently for ~7 days. During this period, no sample was placed inside the system.

### 2.2. Collagen-Based Membrane Production

#### 2.2.1. Collagen Gel Preparation

Collagen type 1 from bovine tail (Gibco) was used. CVM production was based on methods described previously [18,19]. Advanced Dulbecco’s Modified Eagle’s medium (DMEM) (Gibco–Thermo Fisher Scientific, Grand Island, NY, USA) was prepared with 1% penicillin–streptomycin (Gibco) and 8% qualified fetal bovine serum (Gibco) and kept on ice. Next, 22 mM HEPES (Gibco) was added and mixed with a cold type-I collagen solution (5 mg/mL) at a 1:1 ratio until a uniform yellow mixture was obtained by gentle pipette resuspension. The pipettes were also kept on ice to avoid early gelation. Twelve-well plates were used as casts for 2.2 mL of the collagen mix (2.8 μL/mm^2^ of collagen; 1x volume). Gel casts were also prepared at volumes of 2X and 3X collagen. The plates containing the collagen mix were placed in an MCO-18AIC incubator (Sanyo, Osaka, Japan) and kept at 37 °C in 5% CO_2_ for 2 h until gelation. 

#### 2.2.2. Desiccation

The plates containing collagen gels were placed inside the matryoshka system. The temperature and shaking speed of the incubator were set at 40 °C and 30 rpm, respectively. The system remained closed, and T and RH were registered daily. Around day 10–13, the volume of the collagen gels decreased, and membranes were formed. The matryoshka system was then opened and the membranes were rinsed with bi distilled water until the elimination of the phenol red in the medium. The samples were placed back inside the system to complete a period of 37 days of incubation.

#### 2.2.3. Membrane Re Hydration

The plates were removed from the matryoshka system. Bi distilled water was poured into the wells of the plates containing the collagen membranes, left for ~20 min, and then removed. The borders of each membrane were lifted using water pressure. Rounded-tip forceps were used to gently pull out the membranes from the bottom of the well.

### 2.3. Collagen-Based Membrane Characterization

#### 2.3.1. Optical Microscopy

Membranes were humidified for 20 min and the water excess was removed. The surface of CVM was observed with an Axiovert 40 CFL contrast microscope (Carl Zeiss Microscopy, Jena, Germany) and photographed. 

#### 2.3.2. Scanning Electron Microscopy (SEM) 

Samples of the membranes were coated with gold using a Quorum QR150 ES sputtering system (Quorum Technologies, Laughton, UK). The surface and transversal ultrastructure were observed and measured with an EVO MA Scanning Electron Microscope (Carl Zeiss Microscopy). The fibers of samples that were desiccated for 7 days (4 samples, 34 fibers per sample) were photographed and measured using NIH ImageJ. Briefly, image calibration to microns was performed, followed by the selection of linear regions of interest and accumulation using the *Analysis>>Measure* route in ImageJ software.

#### 2.3.3. D Confocal Microscopy

Surface regularity and sample thickness were assessed in triplicate using an Axio-CSM 700 50× objective (Carl Zeiss Microscopy). The samples were hydrated for 20 min prior to data collection.

#### 2.3.4. In Vitro Cell Adherence, Viability, and Cytotoxicity

The NIH3T3 cell line was obtained from ATCC (CRL-1658^™^). The CVMs were cut into circles of 5 mm diameter to fit a 96-well plate. The CVMs were sterilized with Microdacyn (Oculus Innovative Sciences, CA, USA) and rinsed with sterile water. The water was removed and the membranes were left to dry and adhere to the bottom of the wells. NIH3T3 cells were seeded (~10,000) in 100 µL of DMEM F12 (12491-015; Gibco) and incubated overnight at 37 °C in 5% CO_2_.

Cytotoxicity and viability tests were performed using Cell Titer Blue (Promega, Madison, WI, USA) according to the manufacturer’s instructions. Briefly, 20 µL of Cell Titer Blue was added to each experimental, assay control, and sample control well, followed by gentle resuspension and incubation at 37 °C in 5% CO_2_ for 2 h. The same conditions were used to perform a 2–50 × 10^3^ cell fluorescence ladder comparison. Fluorescence was recorded on a Synergy HT spectrophotometer (BioTek, Winooski, VT, USA). Triplicates were tested for each membrane concentration in the experimental condition (membrane + cells), the sample control condition (cells alone), the negative control condition (membrane alone), and the assay control condition (medium + Cell Titer Blue).

#### 2.3.5. Fourier-Transformed Infrared Spectra (FTIR)

Functional groups were identified on each membrane sample using an infrared spectrophotometer coupled with a Fourier transform Spectrum 400 apparatus (Perkin Elmer, Waltham, MA, USA) and recorded in the wavenumber range of 4000–400 cm^−1^ at room conditions.

#### 2.3.6. Transmittance Analysis

Samples of 1X, 2X, and 3X membranes were fully hydrated using bi distilled water and placed into clear cuvettes. Absorbance was acquired throughout the UV–VIS spectra (380–700 nm) using a Synergy HT spectrophotometer (BioTek, Winooski, VT, USA) and transformed to transmittance using the Beer–Lambert law equation. Experiments were performed in triplicate.

#### 2.3.7. X-ray Diffraction Analysis (XRD)

XRD was used to assess the crystallinity of the CVM. XRD was recorded in the 2θ range between 10° and 85° with a step size of 0.026 using a PANalytical Empyrean diffractometer (PANalytical, Almelo, The Netherlands) and CuKα radiation (λ = 1.5406 Å). The voltage applied was 45 kV and the current was 40 mA. 

#### 2.3.8. Ex Vivo Surgical Manipulation Test

To test the ease of surgically manipulating the CVM before performing transplants in the animal models, an ex vivo test was used. Cow eyes from a local butcher shop were fixated over styrofoam plates. All three CVM types (1X, 2X, and 3X) were stained with Trypan blue by 1 min immersion and transplanted into the anterior chamber, which is similar to a Descemet’s stripping endothelial keratoplasty—the surgical procedure for corneal endothelium transplantation. The surgical procedures were performed by two ophthalmic surgeons in order to determine whether the CVM: (1) resisted manipulation with common surgical instruments in this type of surgery to avoiding breakage, (2) was able to introduce them folded through the incision in the periphery of the cornea, and (3) could expand once inside the anterior chamber. Appendix A demonstrates the transplantation of a membrane in the ex vivo model.

### 2.4. Engineered Corneal Endothelium Assembling

We previously reported the ability of CECs to adhere to and proliferate over CVMs [32]. We assembled the CECs/CVM construct for further transplantation to determine its ability to adhere to the corneal stroma and to restore clarity in a corneal opacity model.

This study was approved by the institutional local ethics committee (School of Medicine of Tecnologico de Monterrey), number 2019-003. All the animals were treated according to the Guide for the Care and Use of Laboratory Animals, adhering to the guidelines for the human treatment and ethical use of animals for vision research stated by the National Institutes of Health guide for the care and use of Laboratory animals. White New Zealand rabbits were housed in individual cages and fed ad libitum.

Two 3-month-old male White New Zealand rabbits were used to isolate CECs according to a previous methodology [32,33]. Briefly, White New Zealand rabbits were euthanized with an intravenous pentobarbital lethal dose (90–180 mg/Kg). The endothelium was surgically detached from the corneal stroma. A digestion was made under sterile conditions in a flow hood with 1 mg/mL of collagenase type I (Sigma-Aldrich Co., St. Louis, MO, USA) at 37 °C for 1 h. The CECs were cultured using OptiMEM-I supplemented with 8% FBS, 20 ng/mL of nerve growth factor (NGF; Sigma-Aldrich Co.), 5 ng/mL of epidermal growth factor (EGF; Sigma-Aldrich Co.), 200 mg/L of calcium chloride (Sigma-Aldrich Co.), 20 μg/mL of ascorbic acid (Sigma-Aldrich Co.), 0.08% chondroitin sulfate (Sigma-Aldrich Co.), and 1% antibiotics until confluency. A passage was conducted, and the CECs were cultured using basal media OptiMEM-I supplemented with 8% FBS and 1% antibiotics until confluency. A second passage was carried out, and the CECs were seeded over 2X-CVM (according to the results of the ex vivo model) at a density of 2500 cells/mm^2^ overnight. The CECs/CVM construct was analyzed using light microscopy to register the cell polygonal morphology and cell adherence.

### 2.5. Pilot Study of the Transplantation of Engineered Corneal Endothelium in an Animal Model

In a previous study, we demonstrated the in vivo biocompatibility of CVM in young rabbits [34]. We transplanted the CECs/CVM construct into old rabbits to determine its potential to promote corneal endothelium healing. Five 20-month-old male White New Zealand rabbits were used for the corneal damage model, given that they do not restore corneal clarity after damage according to a previous study [35]. Local and general anesthesia were applied using tetracaine (Ponti Ofteno, Laboratorios Sofía, Jalisco, Mexico) and intramuscular 30 mg/Kg ketamine and 5 mg/Kg xylazine, respectively. A paracenteses was created, and an anterior chamber maintainer was introduced for the injection of 1.2% hyaluronic acid to promote the adhesion of the CECs/CVM construct. The Descemet membrane, along with the corneal endothelium, was surgically removed. After 3 to 5 min, a central corneal opacity was developed and the transplantation of the CECs/CVM constructs was carried out in 4 eyes (experimental group). Appendix A demonstrates the transplantation of the construct. Three eyes were transplanted using only the collagen membrane (control group). Post-surgical analgesic and antibiotic schemes were conducted using topical dexamethasone (Soldrin, Pisa, Guadalajara, Jalisco) 1 mg/mL, 1 to 2 drops every 8 h for 48h, subcutaneous flunixin (Sanfer, CDMX, Mexico) 1–2 mg/Kg every 12 h for 3 days, and intravenous enrofloxacin (Bioquin, BioZoo, Jalisco, Mexico) 5–10 mg/Kg every 8 h for 7 days. Clinical follow-up and photodocumentation was conducted daily for 90 days. The rabbits were euthanized with a pentobarbital lethal dose. The corneas were excised and analyzed by light microscopy. 

The corneal endothelium was detached and analyzed by immunocytochemistry to analyze the presence of ZO-1 and NA/K-ATPase, molecular markers of CECs. Briefly, the corneal endothelium was fixed with cold 4 °C methanol for 24 h and rinsed 3 times with PBS. Unspecific epitopes were blocked with 5% bovine serum albumin (BSA) (Sigma-Aldrich, Co.) at 37 °C for 30 min. Primary antibodies, rabbit polyclonal for ZO-1 (Invitrogen, Waltham, MA, USA) 1:100, and mouse monoclonal anti-alpha 1 Na-K/ATPase (Abcam, Cambridge, MA, USA) 1:100 were used. The secondary antibody Alexa 488 (2:500) (Abcam) was used for ZO-1 primary antibody, and Alexa 568 (1:500) (Abcam) for Na-K/ATPase primary antibody. DAPI (Sigma-Aldrich Co.) was used as a nuclei counterstain. The images were analyzed using ImageJ software [36].

#### Statistical Analysis

Statistical comparisons of the experimental groups were carried out using paired Student’s *t*-tests or analysis of variance (ANOVA). The significance was set at *p* < 0.05. Microsoft Excel (2013; Redmond, WA, USA) and Systat Sigma Plot (V. 11; San Jose, CA, USA) were used for data processing, statistical analysis, and graph generation.

## 3. Results

### 3.1. Matryoshka System Assembling and Stabilization

The temperature inside the system remained constant at 40 °C (SD of ± 0.19 °C at the stabilization phase and SD of ± 0.64 °C in the presence of the samples). The RH at the stabilization phase ranged from 47% to 42% (SD ±1.68%), whereas it ranged from 51% to 33% (SD ± 5.52%) during the sample-desiccation period (Figure 2). A volume of 50 mL of the saturated salt solution lasted for ~100 days when the samples were inside the matryoshka system.

### 3.2. Membrane Desiccation and Rehydration

The formation of the membranes occurred when the thickness of the collagen decreased to ~1 mm (Figure 3). Gels at 1X collagen concentration required 10–13 days to reach the membrane state, whereas 2X and 3X required approximately 15 and 20 days, respectively. We also observed that phenol red must be removed during the gel-to-membrane transformation to avoid further interference with transparency. At the end of the 37-day desiccation period, the membranes were completely transparent. Membrane rehydration for ~20 min was necessary to achieve a good retrieval from the cast. A malleable, transparent material was observed in water suspension. The membranes were placed in a water drop on plastic film, the borders were unfolded, and water was removed using a pipette to straighten the membranes, which were left to dry again at room temperature.

### 3.3. Confocal, Optical, and SEM Characterization

A homogeneous surface was observed after 37 days of desiccation in the three membrane types (Figure 4 panel I a). The collagen fibers were observed at day 7 of desiccation with an average diameter of 1.3 µm ± 0.23 (Figure 4 panel I b). The fibers were not visible at day 37. Confocal microscopy was performed at a thickness of 3.65 μm ± 0.8 for 1X collagen membranes, 4.8 μm ± 0.04 for 2X membranes, and 7.2 μm ± 0.35 for 3X membranes (Figure 4 panel II). the Membrane thickness increased with the collagen concentration and lamina density (Figure 4d–f). 

### 3.4. Membrane Characterization: Cell Viability, IR Spectra, Transmittance, and X-ray Diffraction

NIH3T3 mouse fibroblasts showed adherence to 1X, 2X, and 3X CVM (Figure 5). A fluorescence analysis indicated that the 10 × 10^3^ cells that were seeded initially proliferated to ~31.7 × 10^3^ cells in wells with no membranes. The populations of cells cultured on 1X, 2X, and 3X CVM increased to ~35.8 × 10^3^ ± 5623, ~29.6 × 10^3^ ± 2577, and 29.5 × 10^3^ ± 939 cells, respectively. The cell populations increased threefold within 48 h. One-way ANOVA detected no significant differences between the means of all four samples (*p* = 0.323, n = 3). However, compared with controls, the populations of cells cultured on 1X CVM increased by 12%.

FTIR studies shows characteristic collagen spectra. The band at 3281 cm^−1^ (Figure 5, II IR spectra, arrow 1), indicating the formation of O–H bonds, is smaller in 3X collagen membranes than in 1X and 2X membranes. Moreover, the fingerprint area corroborated the identity of collagen in the membranes as follows. The Bands at 1660 (Amide I band), 1627, 1635 (β-sheet 2ry structures of Amide I)**,** 1637 (triple helix), and 1679 cm^−1^ (stretching C=O vibrations that are H bonded) were fused into a peak with the least transmittance at 1635 cm^−1^ (Figure 4, IR spectra, arrow 2). The band at 1635 cm^−1^ indicating β-sheet secondary structures was correlated with the laminar structure shown in transversal SEM sections. The band at 1535 cm^−1^ (Figure 4, IR spectra, arrow 3) indicated Amide II, whereas the band at 1240 cm^−1^ (Figure 4, IR spectra, arrow 6) was characteristic of collagen. The contribution of bands at 1446, 1396, and 1202 cm^−1^ is seldom (if at all) reported in the literature (Figure 5, II IR spectra, arrows 4, 5, and 7). The bands at 1084 and 1029 cm^−1^ (Figure 4, IR spectra, arrows 8 and 9) are related to nucleic acids. The band at 1084 cm^−1^ corresponds to the phosphodiester bonds of the nucleic acid phosphate/sugar backbone. In contrast, the vibration at 1029 cm^−1^ corresponds to collagen and the phosphodiester groups of nucleic acids. Finally, the vibrations at 565 and 527 cm^−1^ (Figure 5, II IR spectra, arrow 10) correspond to phenyl group torsion. All the bands were intensified as the collagen volume increased.

The membranes produced using our method were ~90% transparent in the visible light spectrum (Figure 5, III Transmittance). The average transmittance for 1X membranes was 92.6% ± 4.91, whereas it was 94% ± 4.40 for 2X and 89.21% ± 4.80 for 3X membranes. Lower transmittance values were recorded at violet wavelengths (ranging from 380 to 450 nm): 1X CVM, 80.4%; 2X CVM, 82.98%; and 3X CVM, 77.56%. Moreover, 1X CVM yielded a transmittance peak of 99.6% at 560 nm and 2X CVM had a peak of 99% at 540 nm, both of which are in the green wavelength spectrum. In contrast, 3X CVM exhibited no peak but yielded higher transmittance values (94.7%) at 650 nm. Percent transmittance increased progressively and stabilized at 510 nm (1X) and 590 nm (2X and 3X). Increasing the collagen concentration decreased the transmittance by ~2.5% per added volume, with 1X CVM showing the highest percent transmittance. ANOVA indicated the presence of significant differences (*p* ≤ 0.001) between the 3X CVM and the 1X and 2X CVMs.

The XRD patterns of pure collagen type 1 (1X, 2X, and 3X CVM) are shown in Figure 5, IV (X-ray diffraction). All the samples exhibited a typical broad hump around 22°, indicating that the collagen was in an amorphous phase [37,38]. A high-intensity peak was observed at 25.9°, which was related to the hydroxyapatite (HA) 002 plane. A shoulder of the HA 320 plane in the hump was shown at 39.8° compared with datasheet 9-432 from the Joint Committee of Powder Diffraction Standards. All samples showed a similar pattern regardless of the content of collagen.

### 3.5. Wet Lab with Ex Vivo Model for Surgical Manipulation Test

The 1X CVM did not resist surgical manipulation for folding of introduction through the corneal peripheral incision. It was fragile and fragmented easily. The 2X and 3X CVM resisted surgical manipulation for the folding and introduction into the anterior chamber, and they were also manipulable for unfolding once inside the anterior chamber (Figure 6). Appendix A shows the implantation of a 2X membrane into the ex vivo model. Taken together with the cell adherence results, the 2X membranes were selected for further construct assembling and pilot transplantation.

### 3.6. Engineered Corneal Endothelium

After overnight incubation, there were confluent zones of CEC over the 2X CVM (Figure 7). However, the overall confluence was ~30% in all the membranes. Given that prior analysis demonstrated that the membrane allows cell proliferation, the transplantation of the constructs to the animal model was conducted to evaluate the ability of the CECs to proliferate in vivo over the membrane and to restore corneal clarity.

### 3.7. Pilot Transplantation 

All eyes (7) that underwent surgical removal of the corneal endothelium developed corneal opacity after 5 min. The four eyes transplanted with CECs/CVM constructs (experimental group) showed peripheral corneal clarity and central opacity after 3 weeks, which was maintained consistently for the 90-day follow-up. The three eyes transplanted with CVM (control group) showed full corneal opacity during the 4-week follow-up (Figure 8). No degradation of the CVM was observed.

The excised corneas transplanted with construct or CVM showed that the membrane adhered to the stroma. The endothelium of the corneas transplanted with construct showed polygonal cells positive for ZO-1 and Na-K/ATPase by immunocytochemistry analysis throughout the tissue. Light microscopy showed polygonal cells all over the collagen membranes of eyes transplanted with the constructs (Figure 9). No cells were observed in the eyes transplanted with the collagen membranes alone. 

## 4. Discussion

Collagen membranes have potential uses in corneal engineering [10,12,19,20,39,40,41,42]. The methods reported for producing them include blending and crosslinking, air-drying and freeze drying, and vitrification [18,19,20,39,40,42,43,44]. However, multiple fabrication parameters result in a lack of consistency of the resulting material properties, limiting their potential for clinical applications. Here, we present a standardized method to fabricate membranes based on previously reported parameters of collagen type, source, and concentration at 40 °C and 40% RH [18,19,20]. We increased the desiccation time, as previous reports did not yield functional membranes. We also tested three collagen concentrations, as this has an effect on the mechanical properties and pattern of fiber distribution, which in turn impact cell adhesion and migration [17].

We used a matryoshka desiccation system that provided a stable RH and enabled us to carry out the vitrification process. The RH variation was similar to that reported for commercial humidity chambers. The desiccation time yielded transparent membranes that were easy to remove from the plates.

The thickness of the CVM relied on the time of desiccation more than on the collagen concentration [19]. Overall, our membrane’s thickness resembled that of the human corneal endothelium [45] and was 3 to 16 times thinner than that reported previously using collagen [18,20] and other materials such as amniotic membrane, human collagen, silk fibroin, hyaluronic acid, and decellularized extracellular matrix, with a thickness of 20–50 µm [40,46,47]. Our SEM data showed randomly arranged fibrils at the surface of the membranes after short desiccation periods. The fibers observed here were larger by an order of magnitude compared with the sizes of collagen membranes reported previously and resembled the early fibrils reported for collagen gels [19,20,38]. This is attributed to the different order of the neutralization and blending steps used for gel formation [48]. Moreover, the incubation conditions cause an increase in the thickness of fibrils: the lateral fusion of discrete (~4 nm) subunits leads to an increase in fibril diameter, followed by longitudinal growth [40]. Fibril bundling and total unification (greater fibril density and homogeneity) were observed when the drying time was increased and high temperatures were used, leading to the production of the homogeneous and smooth surface observed in 1X, 2X, and 3X CVM. A similar non-fibrillar surface was reported in collagen membranes crosslinked with 1-ethyl-3-(3-dimethyl aminopropyl) carbodiimide (EDC), N-hydroxysuccinimide (NHS), and UV treatment [37,49].

T, RH, and time of desiccation impact CVMs’ mechanical strength. Using similar T and RH, Calderon-Colon reported a tensile strength range of 0.69–8.69 MPa. A preliminary analysis of the mechanical behavior of our CVM showed higher strength (9.4 MPa) in the case of the 3X CVM (Appendix A) [20,37].

The transmittance of our membranes was higher than that of the human cornea and 10–12% higher than that reported for similar membranes [19,39,41,42,50,51]. The increase in collagen deposition (i.e., membrane thickness) sets a light transmittance threshold and pattern [51]. Therefore, the tailoring of the ultrastructure of these membranes by changing desiccation conditions and collagen concentration allows for the convenient adaptation of their optical transmittance. 

FTIR and X-ray diffraction analyses are not often provided in the characterization analysis of this type of membrane. However, they are useful in the demonstration of the crosslinking level, data on the origin of the collagen, and desirable properties regarding biological responses, such as enhancement of mineralization, long-term degradation, and tissue integration [38,48,52]. 

The three types of CVM allowed for cell adhesion and proliferation [53]. A slightly reduced adhesion ability was observed in the 2X and 3X CVMs, with no significant differences. Contrasting results have been reported in similar membranes: adhesion and proliferation of different corneal cell types (with no data analysis) [18,20,39] and the need for centrifugation and fibrin to enhance cell adhesion [10,40]. Further surgical manipulation in the ex vivo model allowed us to identify the most suitable sample for transplantation purposes. We chose 2X CVMs for transplantation given that they allowed CEC adherence and were easy to surgically manipulate. Previous studies have reported similar results with tests in suturing corneal tissue [39,41]. In addition, this type of wet lab is reported to be a cost-effective technique for the acquisition of surgical abilities in engineered corneal transplantation [54,55]. 

The transplantation of the CVM/CECs constructs in our pilot study showed that our CVM is suitable for surgical manipulation, adheres to the corneal stroma, and allows CEC proliferation in a model of corneal opacity with high similarities to the human ocular physiology [33]. The transparency was partially restored, even though the cell confluence of the CVM was not full. The effectiveness of corneal endothelium restoration has been proven for collagen membranes fabricated from porcine atelocollagen and bone human collagen [40,54]. Commercially available collagen membranes have shown contrasting results: low adherence to the corneal stroma with low cell viability after transplantation [52] and full recovery of corneal clarity [8]. Similar CVMs have been used only for corneal epithelium healing [37,38].

Although CECs proliferated and covered all of the CVM area, a fibrosis process might take place in the tissue more quickly than the CE can recover [8]. As a pilot study, the transplantation demonstrated for the first time the potential of our CVM for corneal endothelium engineering [55].

## 5. Conclusions

Here, we present a detailed and reproducible method for producing CVM for corneal engineering. The optimal set of T, RH, and desiccation time conditions in a matryoshka assemblage enabled improvements in transmittance, thickness, fiber size, and promotion of cell adhesion. The CVM produced by this method proved for the first time to have the potential for corneal endothelium healing in a preclinical model. Additionally, we provide a customizable method to produce membranes with potential clinical uses in different ocular sites.

## Figures and Tables

**Figure 1 polymers-14-03556-f001:**
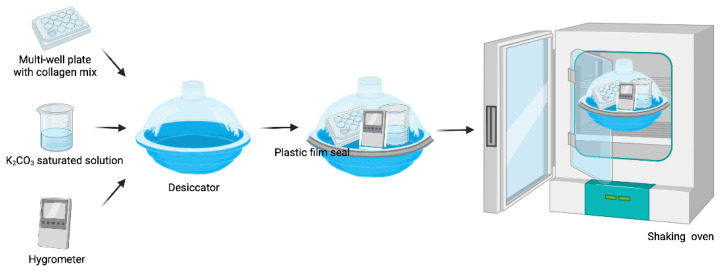
Matryoshka system assembly (created with Biorender.com Available online: https://app.biorender.com (accessed on 21 June 2022).

**Figure 2 polymers-14-03556-f002:**
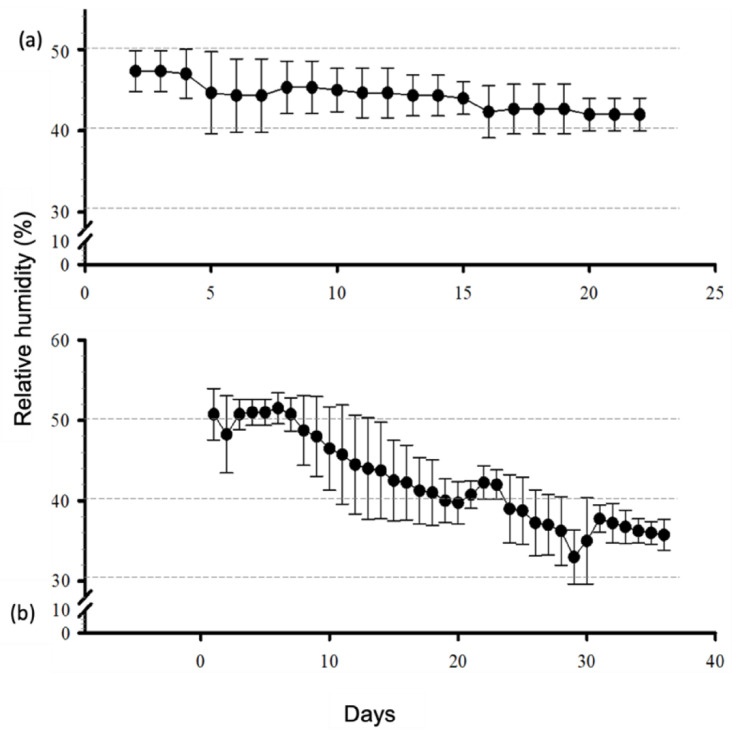
Daily behavior of the relative humidity percentage at the stabilization phase (**a**) and in the presence of 6 mL of collagen gel (**b**).

**Figure 3 polymers-14-03556-f003:**
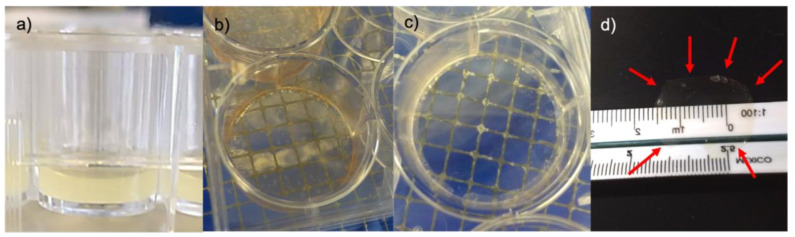
Desiccation process: (**a**) gel before vitrification; (**b**) membrane with phenol red; (**c**) rinsed membrane at day 37; and (**d**) membrane removed from the cast and dried at room temperature.

**Figure 4 polymers-14-03556-f004:**
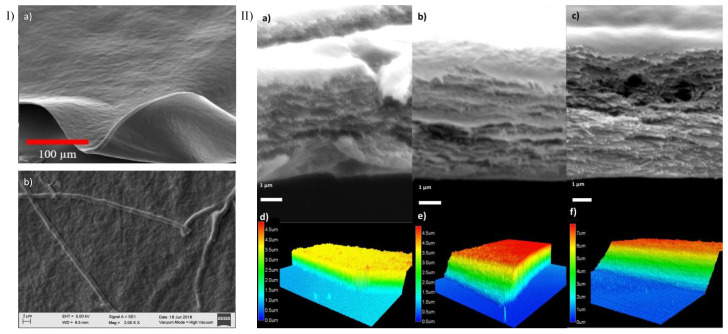
Representative surface and thickness evaluation using material confocal microscopy. Panel (**I**) (**a**) Surface micrograph of a 37-day desiccated 1X membrane. (**b**) Collagen fiber on the surface of a 1X CVM at day 7 of desiccation. Panel (**II**): SEM transversal-view micrograph of a 1X (**a**), 2X (**b**), and 3X (**c**) CVM in wet conditions. (**d**) 1X, (**e**) 2X, and (**f**) 3X CVM showing a thickness of ~3.6, 4.8, and 7.2 μm, respectively.

**Figure 5 polymers-14-03556-f005:**
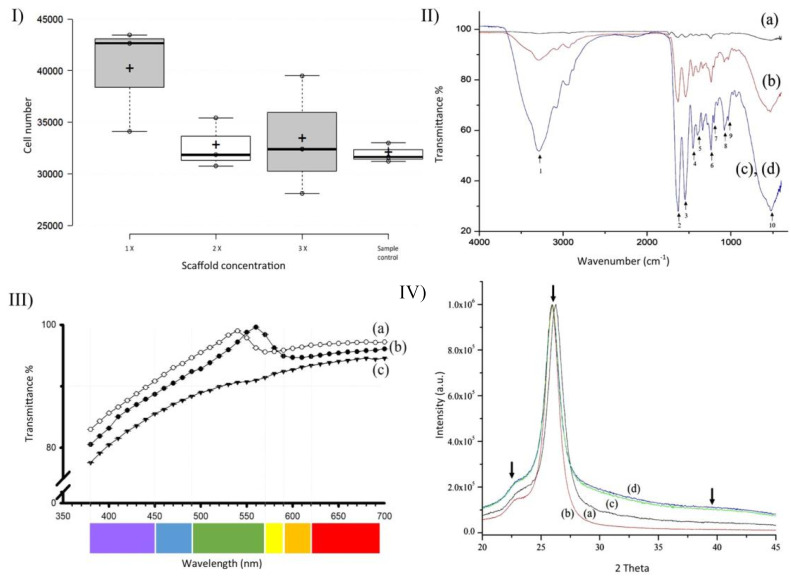
(**I**) **Cell adherence.** NIH3T3 cell count at 48 h when cultured over 1X, 2X, and 3X collagen membranes. Sample control lacks collagen membrane. (**II**) **IR spectra** of pure collagen type 1 (a), 1X collagen membranes (b), 2X collagen membranes (c), and 3X collagen membranes (d). (**III**) **Transmittance**. Percent of transmittance of membranes fabricated using varying collagen volumes: 1X (a), 2X (b), and 3X membranes (c). (**IV**) **X-ray diffraction.** Patterns of pure collagen type 1 (a), 1X (b), 2X (c), and 3X collagen (d) membranes. A 0.6° offset was introduced to detect each pattern clearly. The arrows indicate the 22°, 25.9°, and 39.8° humps for amorphous collagen and the two hydroxyapatite planes.

**Figure 6 polymers-14-03556-f006:**
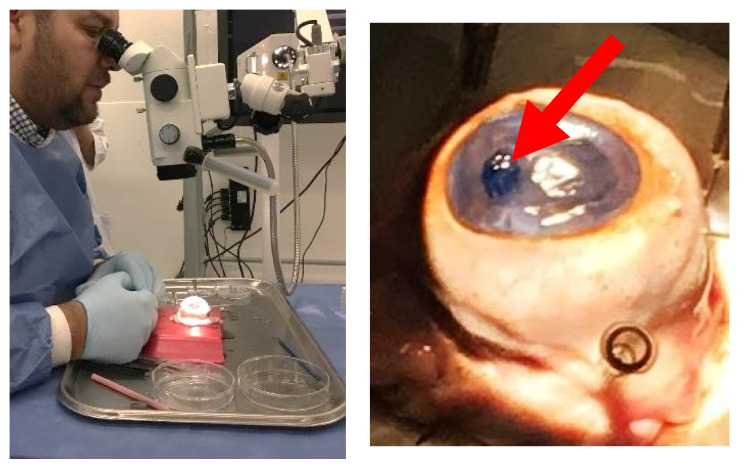
Ex vivo model for surgical manipulation test of collagen membranes. Manipulation of the model under the stereoscope (**left**). Membrane dyed with trypan blue transplanted into the de anterior chamber of the ex vivo model (**right**).

**Figure 7 polymers-14-03556-f007:**
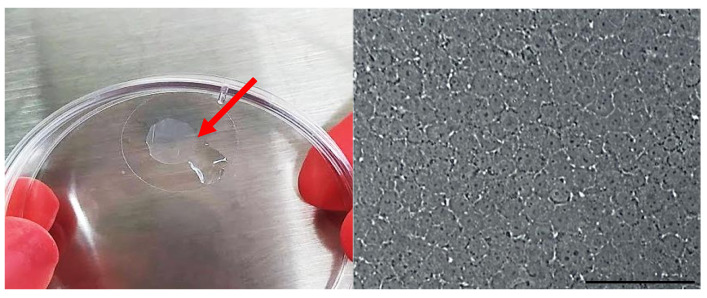
Engineered corneal endothelium (arrow): construct made of CVM and CECs (**left**). Microscopic view of CECs cultured over 2X CVM (**right**). Scale bar = 100 μm.

**Figure 8 polymers-14-03556-f008:**
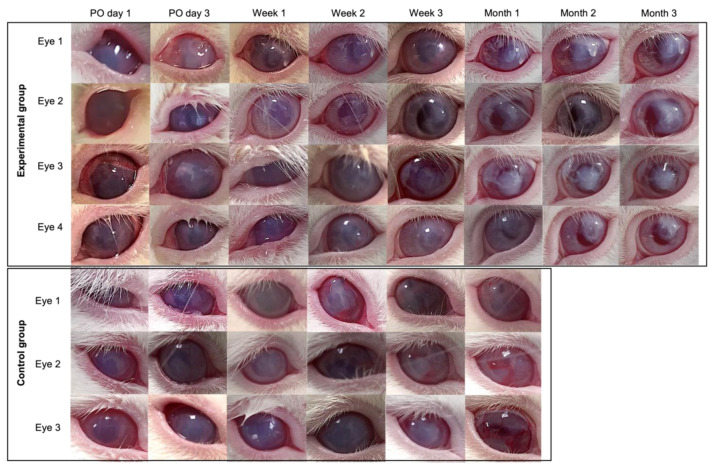
Postoperative follow up of eyes transplanted with CECs/CVM construct and with 2X CVM alone. The eyes with CECs/CVM transplant recovered peripheral transparency, while eyes transplanted with 2X CVM consistently showed edema.

**Figure 9 polymers-14-03556-f009:**
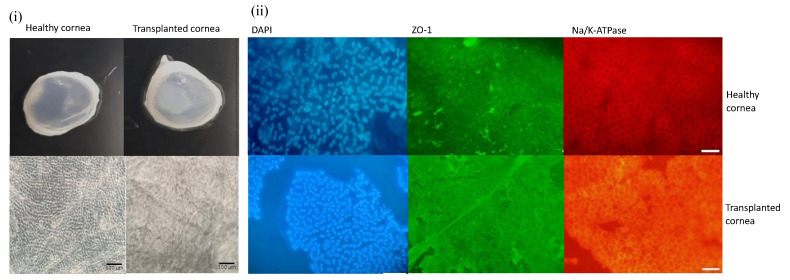
(**i**) Macroscopic view of representative healthy and transplanted corneas and phase contrast microscopy showing corneal endothelium. Scale bar = 100 μm. (**ii**) Immunocytochemistry analysis showing the presence of the ZO-1 and Na/k-ATPase markers in the corneal endothelium of a healthy and transplanted cornea. Scale bar = 20 μm.

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
