# Peer review of "Customizable Collagen Vitrigel Membranes and Preliminary Results in Corneal Engineering"

_polymers, 2022, doi:10.3390/polym14173556_

Round 1

Reviewer 1 Report

The manuscript by Montalvo-Parra et al. entitled, "Customizable collagen vitrigel scaffolds and preliminary results in corneal engineering" describes a method of producing collagen membrane which can be used for corneal tissue engineering. The authors produced collagen membrane by a simple drying method (using matryoshka system). The membranes of different thicknesses were produced by changing the initial volume. The authors provided confocal and SEM images to show the micro/nanotopography of the samples, FT-IR and XRD to show the chemical and physical structures, cell compatibility by culture of NIH 3T3 cells, an ex vivo and in vivo study to demonstrate its usefulness in corneal tissue engineering.

Overall, the manuscript is very poorly written. It does not provide enough details and contexts of their experimental procedures, and therefore, it is very difficult to understand why/how certain experiments were performed (e.g. ex vivo and in vivo experiments). The data analysis for in vitro, ex vivo and in vivo experiments are very shallow. Considering these, it is NOT recommended that this manuscript be published in the journal Polymers.

 Below is some additional critique of the manuscript.

 1. Ex vivo test. What was the purpose of ex vivo test? Which procedure was performed, how, and why? The method section only mentions, "Descemet's stripping endothelial keratoplasty surgical procedures". No added explanation. What was the conclusion from the ex vivo test? What observations were made to come to that conclusion?

 2. Engineered corneal endothelium. No context or background information was provided. Were endothelial cells added to the membrane? If so, what was the initial seeding density? And what was the purpose of this experiment?

 3. In vivo. What is Descemethorrexis? Was there any non-treatment group? Was there any reason that histology was not performed? Was there any degradation of collagen over the 3 month period?

 4. It is not clear how the method described in this manuscript is an improvement over other existing methods.

 5. Some sections numbers are entirely missing: 3.2 – 3.7

 6. The authors mention "width". Do they mean thickness? It is not easy to understand what width means and why it is an important parameter to discuss.

Reviewer 2 Report

In this mannuscript, the presented a detailed and facile method to produce CVM scaffold and further demonstrated it potential application in tissue engineering. Remarkably, very detailed experiment design and protocols are provided, which make it easy for other researchers to follow up this study. I would recommend accepting this manuscript after minor revision:

1. In “Confocal, optical, and SEM characterization” section, the author claimed, “The average fiber diameter was 1.3 μm (SD ± 0.23)”. However, I can not see any supports on this statement from Figure 3. Could the author clarify this?

2. Mechanical strength is usually a very important factor for scaffold biomaterial. Mechanical test on produced CVM scaffold should be a big plus for this work.

Overall, excellent work!

Author Response

We appreciate the revision to our document. We made the changes and answered the suggestions of the Editor and Reviewers. We consider they substantially improved our manuscript quality. The point-by-point answers are described below.

  1. In “Confocal, optical, and SEM characterization” section, the author claimed, “The average fiber diameter was 1.3 μm (SD ± 0.23)”. However, I can not see any supports on this statement from Figure 3. Could the author clarify this?

Thank you for your observation. We included a representative SEM image of the fiber diameter analysis in the figure. 

  1. Mechanical strength is usually a very important factor for scaffold biomaterial. Mechanical test on produced CVM scaffold should be a big plus for this work.

We appreciate your recommendation. We have preliminary analysis that included simples from CVM 1X and CVM 3X. We included a figure as supplementary material at the end of the manuscript.  

PDF file

We appreciate the corrections suggested along the manuscript in this file.

Title: change “scaffolds” for “membrane”

We changed the term along the manuscript

Introduction

Collagen type I has great potential for tissue. This is not completely true. Other collagen types such as type II or IV other several applications in TE. I suggested rephrasing the sentence to be more accurate

The introduction was re-written, and this sentence was removed.

…promoting cell adhesion and migration Reference missing

We added the missing reference

…water uptake, biodegradability, and in vivo 48 biocompatibility These references are not recent, please update them

The references were updated

Ultra-violet (UV) light treatment and desiccation in specific conditions are 63 used for this purpose, producing vitrified scaffolds.Membranes

The word was changed

Methodology

2.3.7 which were the samples? ~this manuscript has different names for the produced product such as scaffolds or membranes Please be precise and concise. Check the thermology used

Thank you for your observation. We described the samples in this section

Results

3.3 Membrane characterization. P = 0.323, n = 3. Change by lower case

We changed to lower case

Figure 4. I suggested to add some italic letter into the figure and then in the legend. It is confused to read only a) b), c) and d)

.. Example

 I)a or II b

Thank you, we added the italic letters

Transmittance…please check if the a) is 2X instead of 1x

Yes, it was wrong. We changed it

Reviewer 3 Report

Dear Authors,

I suggest that the authors explained better the Matryoshka system, to clarify this thermology (a new figure with a schematic should be considered). Besides that, it should be relevant to show the main contribution of this type of study, namely the main conclusion and the advantage to use in vivo models for this kind of work.

Please see the minor comments in the manuscript.

Author Response

We appreciate the revision to our document. We made the changes and answered the suggestions of the Editor and Reviewers. We consider they substantially improved our manuscript quality. The point-by-point answers are described below.

I suggest that the authors explained better the Matryoshka system, to clarify this thermology (a new figure with a schematic should be considered). Besides that, it should be relevant to show the main contribution of this type of study, namely the main conclusion and the advantage to use in vivo models for this kind of work.

Thank you for your recommendation. We added a figure in the 2.1 section explaining the assembling of the Matryoshka system.

We also stated in the conclusion that the main contribution of this work is setting the optimal conditions to fabricate collagen membranes with a matryoshka system for corneal endothelium engineering. In this study, we provide for the first-time evidence of the potential of vitrified collagen membranes produced with this method for the corneal endothelium restoration. Previous studies have tested them in epithelium or stromal layers.

We hope it is clearer now throughout the document.

Round 2

Reviewer 1 Report

The authors have made many changes to the manuscript. However, those changes did not improve the overall quality of this manuscript. And it is not recommended for publication in Polymers.

The main concern is the shallow data analysis. It is very surprising that no histological analysis was performed after the in vivo experiment. A lot can be learned about inflammation and wound healing process, but no such analysis was attempted. It can't be justified by saying it was a pilot experiment.

The authors did not explain the meaning of ZO-1 and Na-K/ATPase. Why were they imaged in the first place? How were the results utilized to claim that this construct was good for replacing the damaged endothelium?

It is also not clear how the implanted construct remained in place? Was it glued to the tissue after the removal of endothelium? How can the implanted allogeneic cells survive in the host? Was there any immune response?

And the results that are provided in the manuscript are not convincing that the developed construct is improvement over other existing technologies.

Author Response

Thank you for your suggestions. We have answered them in the file attached.

Reviewer 3 Report

I agree with this version

Author Response

We appreciate your suggestions